# Down Syndrome, Obesity, Alzheimer’s Disease, and Cancer: A Brief Review and Hypothesis

**DOI:** 10.3390/brainsci8040053

**Published:** 2018-03-24

**Authors:** Daniel W. Nixon

**Affiliations:** 1Cancer Treatment Centers of America, Southeastern Regional Medical Center, Newnan, GA 30265, USA; dnixonun@aol.com; 2Morehouse School of Medicine, Atlanta, GA 30310, USA

**Keywords:** down syndrome, cancer, Alzheimer’s disease, obesity, leptin, adiponectin

## Abstract

Down syndrome (trisomy 21), a complex mix of physical, mental, and biochemical issues, includes an increased risk of Alzheimer’s disease and childhood leukemia, a decreased risk of other tumors, and a high frequency of overweight/obesity. Certain features related to the third copy of chromosome 21 (which carries the APP gene and several anti-angiogenesis genes) create an environment favorable for Alzheimer’s disease and unfavorable for cancer. This environment may be enhanced by two bioactive compounds from fat cells, leptin, and adiponectin. This paper outlines these fat-related disease mechanisms and suggests new avenues of research to reduce disease risk in Down syndrome.

## 1. Introduction

Unusual and poorly understood relationships exist between Down syndrome (DS), Alzheimer’s disease and cancer. The risk for Alzheimer’s disease and childhood leukemia in DS is increased, but the risk for solid tumors is decreased [1,2]. The observations that individuals with DS are likely to be overweight or obese [3], and that excess body fat is a risk factor for both Alzheimer’s disease and cancer [4], may mechanistically link DS to these two diseases. In people without DS, there is an inverse relationship between risk of Alzheimer’s disease and cancer [5]. Two metabolically active compounds produced by fat cells, leptin and adiponectin, are involved in this inverse relationship [4], and these compounds could play a role in the Alzheimer’s disease cancer risk pattern in DS as well. Leptin has cancer-promoting and Alzheimer’s disease-inhibiting properties, while adiponectin has opposing, cancer-inhibiting and Alzheimer’s disease-promoting ability [4]. This paper reviews epidemiologic associations between DS, Alzheimer’s disease and cancer, the genetic influences of trisomy 21, and how leptin and adiponectin could promote or inhibit these genetic influences. Increased knowledge about these effects could lead to new clinical trials aimed at disease prevention and improved quality of life in people with DS.

## 2. Hypothesis

Excess body fat is common in individuals with DS. Fat cells produce many bioactive adipokines, including leptin and adiponectin. These two compounds have opposing activities that may promote the high rate of Alzheimer’s disease and childhood leukemia and low rate of solid tumors in DS. It is known that leptin levels are increased in young people with DS and adiponectin levels are increased in older individuals. Reduction of abnormal levels of leptin and adiponectin could reduce the risk of childhood leukemia and Alzheimer’s disease. Evidence-based disease prevention clinical trials could focus on weight control and the leptin/adiponectin ratio in individuals with DS.

This article is divided into two Sections for clarity. Section 1 is a brief review of pertinent clinical, etiologic, and epidemiologic features of DS. This Section provides background information helpful in understanding the cellular and molecular interactions described in Section 2. Section 2 is a description of how leptin and adiponectin affect various signaling factors involved in the DS, Alzheimer’s disease, and cancer relationships, and how the underlying genetic consequences of trisomy 21 are involved in these relationships. The article concludes with suggestions for future research in this area.

## 3. Section 1: Clinical, Etiologic and Epidemiologic Features of DS and the Involvement of Fat-Related Adipokines

### 3.1. DS Clinical Features

DS includes a characteristic facial appearance, varying degrees of intellectual ability, low muscle tone in infancy, and an increased risk for many medical problems including infections, pulmonary, thyroid, skin, skeletal, hearing and vision issues, seizures, diabetes, sleep apnea, early menopause, and congenital heart defects [6]. Dr. JLH Down, in his descriptive (but mistaken regarding ethnic classification) 1866 paper [7], added that subjects were “humorous with a lively sense of the ridiculous”. He said that they responded well to treatment including dietary modifications and what today would be speech therapy. Life span was very short at that time but is much longer today. In the mid-1900s, individuals with DS lived about 12 years. Today, with proper medical care, the lifespan is approximately 60 years [6], with some living into their 70′s.

### 3.2. DS Causes

Dr. Down considered several possible causes and concluded that paternal tuberculosis was responsible, but in the 1950s it became clear that DS results from an inherited extra copy of chromosome 21 and is not related to any parental diseases. There are three types of DS. The most common (95% of cases) is a cell division error (nondisjunction) that produces three 21 chromosomes (trisomy 21) in the fertilized egg instead of the normal two. Less common causes include translocation of part of chromosome 21 (3–4% of cases), in which part of chromosome 21 attaches to another chromosome during cell division, and mosaicism (1–2% of cases), when nondisjunction of chromosome 21 occurs after fertilization; some cells then have three and some have two 21 chromosomes [8].

### 3.3. DS and Alzheimer’s Disease

A 1986 review discussed the long-known association between DS and dementia, and that neuropathologic changes resembling Alzheimer’s disease were common in DS patients over the age of 35 years [1]. A later review (2012), found that almost all DS patients over age 40 years have senile plaques and neurofibrillary tangles typical of Alzheimer’s disease [9]. These authors concluded that, in DS, as in the general population, there is a long asymptomatic period (ten years or more) before the onset of dementia. There may be an acceleration in Alzheimer’s disease development after age 40 years, with increased neuropathology along with an increase in concentration of amyloid-beta [9].

### 3.4. DS and Cancer

Children with DS have an increased risk (10–20-fold) of acute lymphoblastic leukemia and acute myelocytic leukemia than children without DS. In sharp contrast, however, solid tumors are much less common in both children and adults with DS [6,10]. A Danish study of over 2800 DS subjects found a 50% lower risk of non-leukemia cancers compared to age-matched controls (24 solid tumors found versus 47.8 expected). No cases of breast cancer were found, but 7.3 were expected. No cases of leukemia occurred after age 29 years [2]. A retrospective British study of hospitalized mentally disabled patients found no cancer cases among 115 DS patients over a 40 year period with 8 expected; the percentage of cancer cases among non-DS subjects increased over the same period [11]. A large tumor registry study in 2016 found a significantly lower risk of all major solid tumor types in DS, including cancers of the lung, breast, and cervix; the risk of testicular germ cell tumors was also increased [12]. A DS tumor profile has been proposed, with hematopoietic and germ cells at increased carcinogenic risk and epithelial, nerve and renal cells at decreased risk [13].

### 3.5. DS, Body Weight and Body Mass Index (BMI)

Most individuals with DS become overweight or obese over time. In a 2017 study, adult DS patients were three times more likely to be overweight/obese than adults without developmental disabilities; 77% of DS subjects in this study had BMI’s in the overweight or obese range with 48.5% obese. In the comparison group with no developmental disabilities, 25.5% were obese [3]. Several mechanisms may be involved in the tendency for weight gain in individuals with DS, but more research is needed. Possible mechanisms include comorbid hypothyroidism, decreased resting metabolic rate, preference for high carbohydrate foods, certain behavioral tendencies that increase with age and high leptin levels; increased leptin levels from leptin resistance correlate with obesity [14]. Unlike current general population trends, the tendency to be overweight/obese in DS is not a very recent development; a 1995 study found that 31% of males and 22% of females with DS were overweight and 48% of males and 47% of females were obese [15].

### 3.6. Body Fat Function: Adipokines

Body fat, especially visceral fat, is now known to be more than just storage of excess energy but is an active producer of hundreds of biologically active compounds called adipokines, including leptin and adiponectin. Both of these adipokines are important in weight control, glucose regulation and hormonal metabolism, and are involved in cancer and Alzheimer’s disease as well. Leptin generally correlates with body fat mass and helps regulate appetite so that food intake decreases as fat stores increase [16]. When body fat increases, however, leptin can fail to control appetite and leptin levels continue to rise (“leptin resistance”) [17]. Leptin is also involved in chronic inflammation and immune responses [18], while adiponectin helps control glucose metabolism, and has anti-inflammatory and cardioprotective activity [19]. Adiponectin levels increase and leptin levels decrease after weight loss; the opposite occurs after weight gain [20,21].

DS and Leptin: In 35 children with DS, compared to unaffected siblings, subjects with DS had higher leptin levels after adjustment for percent body fat; BMI and percent body fat were higher in DS children. Increased leptin resistance, perhaps on a genetic basis, in children with DS was postulated [22]. In a similar study using unaffected, not related controls, leptin levels in DS were increased for percent body fat; leptin was higher in girls than in boys [23]. Higher leptin levels and increased leptin resistance was also seen in a 2017 cross-sectional case-control study of children with DS [24]. Leptin levels were higher in children than in adult and older DS subjects [25]. Increased circulating leptin in obese children with DS was similar to elevated levels in obese children without DS [26].

DS and Adiponectin: In contrast to leptin, whose levels decline with age, serum adiponectin levels were increased in older DS patients compared to children and adults with DS [25]. Fasting adiponectin levels were lower in DS children than in controls, but this did not reach statistical significance [24]. The varying levels of leptin and adiponectin with age may be important in Alzheimer’s disease and cancer pathogenesis.

Leptin and Alzheimer’s Disease: Framingham Heart Study data showed a prospective association between high baseline leptin levels and decreased dementia and Alzheimer’s disease later in life. Conversely, subjects with lower baseline leptin levels were at increased risk for later Alzheimer’s disease [27]. Similarly, high leptin levels in the elderly were associated with decreased cognitive decline, and serum leptin levels inversely correlated with dementia [28,29]. Possible reasons for leptin’s anti-Alzheimer’s disease effects include its ability, shown in mice, to reduce beta and gamma secretase activity, decrease cleavage of APP and decrease accumulation of beta amyloid [30,31]. Leptin also reduced the harmful effects of beta amyloid on the hippocampus, protected cortical neurons from beta amyloid related death, and inhibited upregulation of phosphorylated tau [32]. In transgenic mice, leptin treatment reduced beta amyloid accumulation in the brain, decreased toxicity in neuronal cells and, in human neuronal cells, decreased levels of phosphorylated tau [29,33,34,35]. Leptin levels are elevated in children with DS, but levels decrease with age [24,25], supporting the possibility that any protective action of leptin against Alzheimer’s disease declines over time.

Adiponectin and Alzheimer’s disease: One study found low levels of adiponectin in active Alzheimer’s disease patients [36], but two other studies found that increased adiponectin was related to Alzheimer’s disease [37,38]. One (The Framingham Heart Study) showed that increased baseline adiponectin levels were a significant risk factor for later Alzheimer’s disease in women, and another found higher levels of adiponectin in Japanese subjects with Alzheimer’s disease, along with a positive correlation between plasma and cerebrospinal fluid (CSF) adiponectin. Adiponectin also can increase astrocyte inflammation; brain inflammation has been implicated in Alzheimer’s disease [39,40]. In contrast to the pro-Alzheimer’s disease effects of increased adiponectin levels, physiologic adiponectin levels reduced beta-amyloid toxicity in human neuroblastoma cells [41], suggesting a dose-response relationship and the possible importance of the leptin/adiponectin ratio in pathogenic processes.

Leptin and Cancer: Leptin can increase cancer risk and stimulate the aggressiveness of cancer cells, including the obesity-related cancers of the breast and prostate [42,43,44,45]. Potential leptin-related mechanisms in solid tumors include hormonal interactions, angiogenesis, reduction of apoptosis and chronic inflammation. Leptin can increase estrogen sensitivity and progesterone receptor mRNA, stimulate aromatase activity, upregulate vascular endothelial growth factor, activate epidermal growth factor receptor, and upregulate the pro-inflammatory mediators TNF-alpha and IL-1 [46,47,48,49,50]. Leptin signaling can also promote hematologic malignancies [51,52].

Adiponectin and Cancer: In contrast to leptin. adiponectin has several cancer-inhibiting properties. An inverse relationship has been shown for adiponectin levels and breast, endometrial, prostate, colon, esophagus, pancreas, stomach, and myeloid leukemia; adiponectin was inversely associated with stage of disease in stomach, colorectal and prostate cancer [53,54,55,56,57,58]. Possible mechanisms include anti-angiogenesis, induction of apoptosis, activation of tumor-suppressors and activation of cell signaling pathways [53,54,55]. Low levels of adiponectin in obese subjects may increase breast cancer risk via hyperinsulinemia and vascular endothelial growth factor (VEGF) upregulation, and high adiponectin levels may reduce risk of breast cancer [59,60]. Growth of leptin-stimulated prostate cancer cells is inhibited by adiponectin [58]. Adiponectin inhibits growth in myeloid cell lines and induces apoptosis in myelomonocytic leukemia cells [61,62].

### 3.7. Genes on Chromosome 21

The autosomal 21st chromosome is the smallest chromosome but still contains several hundred genes. Overexpression of certain genes due to a third copy of chromosome 21 is thought to underlie the complex components of DS, but the specific genes and the interactions of their protein products are not completely defined. Some of these genes and gene products are known to be involved in aging, immune function, DNA repair and synthesis, hormone signaling, and mental function. Others, pertinent to the DS, Alzheimer’s, and cancer relationship, include the APP gene, transcriptional factors such as ETS proto-oncogene 2 (ETS2), the angiogenesis suppressors Dscr1 and Dyrk1A and the collagen-18 gene whose fragment, endostatin, is an anti-angiogenic compound [63,64,65].

## 4. Section 2: Cellular and Molecular Effects of Leptin and Adiponectin on Alzheimer’s Disease and Cancer Risk in DS

Abnormalities in DS related to the third copy of chromosome 21 include downregulation of DNA repair genes and upregulation of pro-apoptotic and angiogenesis genes. For example, dysregulation of the Notch/Wnt pathway has been associated with premature aging processes in DS that may relate to early Alzheimer’s disease and leukemia [66].

Leptin and adiponectin have definite effects on several of the important genes, gene products and signaling pathways involved in DS, Alzheimer’s disease, and cancer. Alzheimer’s disease has been said to result from neurons being in a “prone to die” state, while cancer cells are “prone to live” [5]. Trisomy 21 creates an environment that tends to accentuate these two states, depending on the age of the person and other unknown influences. For example, increased adiponectin in older DS patients might decrease angiogenesis in concert with the known anti-angiogenesis factors on chromosome 21, leading to increased Alzheimer’s disease risk but reduced adult cancer risk; anti-angiogenesis should not affect any leptin-related childhood leukemia risk, because leukemia does not require new blood vessels to grow.

There is evidence for leptin’s influence on apoptosis, angiogenesis, APP metabolism and the signaling pathways p53, Wnt and Notch. Adiponectin is known to be involved in angiogenesis, apoptosis, APP, p53, and Wnt signaling.

### 4.1. Cellular Effects: Apoptosis and Angiogenesis

Apoptosis, or programmed cell death, is a normal part of tissue growth and development, allowing the removal of old or damaged cells. Abnormalities in apoptosis, however, can lead to either an increase in the “prone to die”, or the “prone to live” state. An increase in the neuronal “prone to die” state, promoting Alzheimer’s disease in DS, may be due in part to altered levels of anti-apoptotic proteins: Bax apoptosis-activating proteins were elevated and anti-apoptotic Bcl-2, heat shock protein 70, neuronal apoptosis inhibitory protein and p53 were decreased [67]. Increased neuronal apoptosis from a defect in metabolism of reactive oxygen species has also been suggested in DS [68]. Obesity, common in DS, is known to induce chronic inflammation and apoptosis, with leptin and adiponectin key players in this process. Leptin and adiponectin can promote or inhibit apoptosis depending on the system studied. Leptin levels are high early in DS, but decline later [24,25]. High levels of leptin and glucose induced pancreatic beta-cell apoptosis through activation of the JNK pathway [69]. This may help account for the increased risk of type 1 diabetes in DS and could initiate early Alzheimer’s disease processes. In cancer, however, leptin promotes the “prone to live” state through anti-apoptotic activity. In liver cancer cells, for example, leptin stimulates proliferation and inhibits apoptosis through crosstalk between several signaling pathways, including cyclin D1 and Bax [70]. Leptin also reduced apoptosis in human endothelial cells, indicating proliferative and proangiogenic activity [71]. In contrast, adiponectin promoted apoptosis and antiangiogenesis through a caspase cascade leading to tumor cell death in mice [72]. Also, adiponectin induced apoptosis in macrophages through reactive oxygen species/reactive nitrogen species generation [73]. In human endometrial cancer cells, adiponectin inhibited cell growth and increased apoptosis through reduction of growth-promoting cell regulators [74], but adiponectin did have an anti-apoptotic effect in pancreatic cancer cells, indicating a tissue-dependent effect [75]. Thus, leptin and adiponectin can have either pro and anti-apoptotic activity depending on the system studied; in cancer, however, leptin has significant anti-apoptotic and adiponectin has pro-apoptotic actions. In Alzheimer’s disease, adiponectin could be promoting through loss of neurons, especially as adiponectin increases with age in DS [24,25]. In DS, lower levels of adiponectin in children might increase leukemia risk because adipontin inhibited myeloid cell proliferation and induction of apoptosis in myelomonocytic leukemia cells [61,62].

Angiogenesis: The late Dr. Judah Folkman proposed an anti-angiogenesis hypothesis in 1971 to explain the low cancer rate in DS [65]. DS subjects have 3 copies of chromosome 21, which carries at least three genes involved in suppressing angiogenesis, a process necessary for solid tumor development. These are Dscr1, Dyrk1A and collagen 18 [65]. A fragment of collagen 18 called endostatin, along with the protein products of Dscr1 and Dyrk1A, suppress angiogenic signaling from vascular endothelial growth factor (VEGF) [65]. Leptin has pro-angiogenic (neovascularization) properties [71], but these may be negated by the three anti-angiogenesis factors on chromosome 21, especially as leptin levels fall with age in DS. On the other hand, leptin’s potential leukemogenic and anti-apoptotic activity [62] should not be impeded by anti-angiogenesis; this could contribute to the increased rate of childhood leukemia in Down syndrome. Adiponectin has anti-angiogenic activity [72], so that high levels in older DS patients [25] might contribute to the risk of Alzheimer’s disease through neuronal blood deprivation. The anti-angiogenic genes on chromosome 21 could add to this effect and counter the pro-angiogenic effects of adiponectin found by others [76,77].

### 4.2. Molecular Effects and Signaling Factors: APP, p53, Wnt, and Notch

Amyloid precursor protein (APP) and the APP secretase cleavage product beta amyloid are thought to be closely involved in Alzheimer’s disease pathogenesis. Chromosome 21 carries the APP gene, so that the three copies of chromosome 21 may be important in the DS and Alzheimer’s disease connection through increased APP and beta amyloid. Beta and gamma secretases cleave APP to create beta amyloid, while alpha secretase cleaves APP in the APP domain creating a neuroprotective product but does not create beta amyloid; protein kinase C and mitogen activated protein kinase regulate alpha secretase cleavage [78]. Leptin regulates beta amyloid damage by down regulation of gamma secretase [31], inhibits tau phosphorylation and beta amyloid production in neuronal cell cultures [29] and helps prevent neuronal cell death due to beta amyloid [32]. Transgenic mice that overexpress APP had low leptin levels associated with hypothalamic dysfunction; low leptin levels are linked to Alzheimer’s disease [27,79]. Leptin, delivered via intra-cerebroventricular injection as HIV-leptin, decreased brain beta amyloid accumulation and toxicity and increased synaptic density [34]. APP and the related amyloid precursor-like protein 2 also have, somewhat surprisingly, significant cancer modulating effects, including increased tumor cell proliferation, migration, and invasion [80]. APP promotes proliferation of breast and prostate cancer cells [81,82]. The APP proliferative effect in prostate cancer may be due to modulation of metalloproteinase genes [82]. The cancer-stimulating activity of APP could act in synergy with similar activities of leptin, and the neuroprotective APP alpha secretase product could synergize with the neuroprotective activity of leptin. Adiponectin is protective against amyloid beta neurotoxicity under oxidative stress at physiologic concentrations [41], but upregulation of adiponectin may increase amyloid deposits; the latter effect may involve oxidative stress and chronic inflammation. Both “gain of function” and ‘loss of function” of adiponectin have been postulated in neuronal cell death [83]. Adiponectin levels rise with age in DS, when Alzheimer’s disease becomes very common [25].

P53 is a gene that regulates apoptosis and acts as a tumor suppressor. In DS, p53 inhibits the Down-associated protein kinase Dyrk1A through increased expression of microRNA miR-1246; enhanced apoptosis results [84]. Activation of p53 can also create a pro-apoptotic integration of several signaling pathways and contribute to Alzheimer’s disease in DS [85]. P53 is known to be upregulated in Alzheimer’s disease [86]. Leptin affects p53 generally in a cell-proliferative, anti-apoptotic direction [87,88]. Leptin stimulates human breast cancer cells in part through down-regulation of p53 [89]. The overall effect of leptin to reduce apoptosis via p53 downregulation could indicate a neuroprotective effect. Adiponectin, in contrast, affects p53 in an enhancing direction that promotes growth arrest and apoptosis. In prostate cancer cells, adiponectin reduced cell inhibition induced by p53 and, in breast cancer cells, adiponectin increased p53 and inhibited cell proliferation [45,90]. Adiponectin can also activate other tumor-inhibiting pathways including MAPK, STAT3, AMPK, mTOR, and NF-κB [20,59]. In summary, leptin and adiponectin have opposing effects on cell growth mediated by p53. The overall effect would increase Alzheimer’s risk in DS and might increase leukemia risk but not solid tumor risk if leptin is counteracted by the antiangiogenesis genes on chromosome 21. Leptin levels decrease with age in DS, which might enhance this counteraction.

The Wnt gene family encodes signaling proteins involved in normal cell processes, but Wnt can also send abnormal signals in cancer and modulate beta-amyloid effects in the brain. In Down syndrome, activated Wnt signaling may be involved in pulmonary hypertension [91]. Dscr5 found in the Down syndrome critical region of chromosome 21 is involved in aspects of embryogenesis through regulation of membrane localization of Wnt receptors [92]. Dyrk1A, also found in the Down syndrome critical region of chromosome 21, is involved in Wnt signaling as well [93].

Leptin and adiponectin both modulate Wnt signaling. Leptin can increase Wnt signaling leading to activation of pathways involving cell proliferation and differentiation [94]. Activation of Wnt and inhibition of negative Wnt regulators decreased beta-amyloid toxicity [95]. Adiponectin can upregulate some Wnt genes but reduce activity of others. Reduced Wnt signaling may increase Alzheimer’s pathology and activation of Wnt may reduce beta-amyloid toxicity; decreased Wnt signaling may thus contribute to Alzheimer’s disease [95,96], and leptin might increase Wnt signaling and be neuroprotective.

Notch signaling is involved in development of multiple mammalian tissues and organs including the hematopoietic, vascular, breast, GI tract and nervous system [97]. Notch is involved in Down syndrome pathogenesis, is disrupted in many cancers, and plays a role in Alzheimer’s disease. The Notch protein is cleaved by gamma-secretase during signaling processes. After gamma-secretase cleavage, the intracellular fragment moves to the nucleus and regulates gene expression. Activation of Notch in Down syndrome may involve cross-talk with APP signaling, which may affect brain development [98]. Gamma-secretase inhibitors impede Notch signaling, resulting in abnormal capillary growth similar to that seen in AD [99]. Dyrk1A decreases Notch signaling which may contribute to abnormal brain development in animal models of Down syndrome [100]. Leptin increases Notch and increased proliferation and upregulation of VEGF in breast cancer cells depends on intact Notch signaling [101,102]. More research is needed on Notch signaling and leptin; little information is available on adiponectin, Notch and DS. The Table 1 summarizes the cellular and molecular mechanisms related to leptin and adiponectin in the DS, Alzheimer’s disease, and cancer relationship.

Several other signaling factors may be involved in DS, Alzheimer’s disease and cancer, but more research is needed. These include ETS, apolipoprotein E (APOE), endostatin and Pin1. The role of diabetes, insulin resistance and chronic inflammation in DS should also receive further attention, as should microRna’s and vitamin D. Functional microRna’s are active in cancer and Alzheimer’s disease [103], and several microRna’s are encoded by genes on chromosome 21 [104]. Subjects with DS are frequently deficient in vitamin D [105], and this vitamin can substantially reduce leptin levels [106].

## 5. Limitations of This Review

The relationships between DS, obesity, Alzheimer’s disease and cancer are very complex. Not enough is known about all the cellular and molecular mechanisms involved to draw firm conclusions. Leptin and adiponectin are just two of hundreds of adipokines produced by fat cells. In addition, many other bioactive compounds change during the pathogenesis of Alzheimer’s disease and cancer. The actions and interactions of these compounds and how they relate to the unique genetic environment in ds require further investigation. In addition, the reasons for overweight/obesity in DS are not fully understood, and conventional exercise-based weight control interventions have not been successful [107]. Another major challenge is the uncertainty of diagnosing dementia in DS, especially as individuals age making accurate correlation with early dementia and adipokines difficult. Conventional dementia screening tools useful in the general population are limited in older people with DS; several ds-specific screening tests have been devised but no rapid screening tool has been validated; early detection of dementia in ds remains elusive [108]. Several radiologic and biochemical tests for early diagnosis of Alzheimer’s Disease dementia exist. These include (11) c-pib-pet [109] and CSF tau and the CSF tau/beta ratio [110]. More experience may lead to reliable diagnostic tests in the clinic and earlier subject entry into clinical treatment trials.

## 6. Discussion

DS (trisomy 21) encompasses a complex and variable spectrum of biochemical, mental, and physical changes related to genetic imbalances from the extra chromosome 21. The DS features important to the present discussion are the high rates of overweight/obesity, the increased risk of childhood leukemia and Alzheimer’s disease and the low risk of solid tumors. Genes on chromosome 21 have influence on these risks, and certain compounds produced by fat cells (leptin and adiponectin) have actions that can add to these risks, either positively or negatively. For example, increased adiponectin in older DS patients could decrease cancer risk already low due to anti-angiogenesis genes on chromosome 21, and at the same time increase Alzheimer’s disease risk. Leptin and adiponectin are both involved in signaling pathways important in cancer and Alzheimer’s disease; these pathways tend to decrease cancer risk (except for childhood leukemia) and to increase Alzheimer’s disease risk, supporting the observation that excess body fat is a risk factor for both cancer and Alzheimer’s disease. The increased rate of childhood leukemia in DS seems to be an exception to the low rates of other cancers, but an interesting possibility is that the anti-angiogenesis genes on chromosome 21 would not have a suppressing role in leukemia, which is not dependent on new blood vessels; any leukemia-stimulating effects of leptin would also not be impeded by decreased angiogenesis. The increased risk for childhood leukemia in DS could therefore be related early cellular mechanisms related to the cancer-promoting effects of high leptin levels from leptin resistance (anti-apoptosis mediated in part through down-regulation of p53), and decreased leukemia—suppressing (apoptotic) effects from lower childhood adiponectin levels. Alzheimer’s Disease risk later in DS could rise as weight increases, leptin levels fall and leptin’s neuroprotective actions diminish. Increasing adiponectin with age and increasing obesity in ds could add to Alzheimer’s Disease risk through inflammatory and pro-apoptotic mechanisms.

Future research directions could include a focus on the leptin/adiponectin ratio over time in DS; leptin is higher in young DS subjects, while adiponectin is higher in older DS subjects. Much more could be learned about the effects of these adipokines on various signaling factors in DS, and the cancer-related interactions of leptin and the anti-angiogenesis factors on chromosome 21. Clinical trials could measure the leptin/adiponectin ratios over time in DS and the effect of weight gain on these ratios. Leptin and adiponectin levels are different in older, overweight/obese DS subjects; leptin is low, and adiponectin is high, but the opposite would be expected with weight gain. Could unknown effects from genetic imbalances in ds account for this unexpected finding? Weight control-based clinical trials could begin when the leptin/adiponectin ratio starts to change or become increasingly abnormal over time. Careful epidemiologic trials could look at diet, exercise, drugs, and other factors in DS weight gain. Also, it is known that low vitamin D is common in DS, and that this vitamin can lower leptin levels. A possible trial could measure vitamin D in DS subjects and follow them over time on vitamin D supplements (if vitamin D is low) to determine the leptin/adiponectin ratio, any change in the ratios and the occurrence of leukemia, solid tumors, and Alzheimer’s disease. Corollary DS studies could look at other disorders, such as skeletal problems (osteoporosis) and hormonal changes on long-term vitamin D supplementation, and effective early dementia diagnosis would allow for evidence-based drug treatment trials. In conclusion, the life span in DS has improved greatly over the last several decades, and well-designed, low- risk clinical trials backed by solid basic research could increase quality of life and life span even more.

## Figures and Tables

**Table 1 brainsci-08-00053-t001:** Cellular and molecular mechanisms involved in the ds, obesity, Alzheimer’s disease and cancer relationships *.

Mechanism	Down Syndrome	Leptin	Adiponectin
Angiogenesis	Decreased	Increased	Decreased
Apoptosis	Increased	Decreased	Increased
APP cleavage	Increased	Decreased	Increased
p53	Increased	Decreased	Increased
Wnt	Increased	Increased	Decreased
Notch	Increased	Increased	Unclear

* The mechanisms affected by leptin and adiponectin tend to reinforce the chromosome 21-related pro Alzheimer’s disease influence. The anti-angiogenesis influence of chromosome 21 would counteract leptin’s pro-angiogenic effect.

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
