# Peer review of "Down Syndrome, Obesity, Alzheimer’s Disease, and Cancer: A Brief Review and Hypothesis"

_brainsci, 2018, doi:10.3390/brainsci8040053_

Round 1

Reviewer 1 Report

Overall, I felt that this paper provided proposed new directions in understanding cancer and Alzheimer’s risk in individuals with Down syndrome. I think that the author tried to take a lot of information and synthesize it into as cohesive a review as possible. I think understanding risk for cancer and alzheimer’s based on metabolically active compounds leptin and adiponectin is interesting and has importance in potential medical treatments.

One of the points I wish would be made clearer is that is sounds like obesity, may cause greater risk for Alzheimer’s disease whereas childhood cancers may be more related to early cellular mechanisms. I feel that needs to be more clearly indicated.

On page 3, when discussing obesity and DS, I’m wondering if the author could discuss etiologies of obesity in DS to help explain why this increases with age.

Some minor points to consider:

Please check punctuation as there are a number of areas where there are incorrectly placed periods or commas.

The table requires some footnote to help it stand alone and explain what it is indicating.

Using person first language “Individuals/peoples with Down syndrome” when speaking of the population

Author Response

Thank you for the helpful review. comments. My responses are:

Comment 1: Obesity and cellular mechanisms.I have added my response in the Discussion section, last page.

Comment 2: Etiology of obesity.  My response is on page 3> 

Comment 4: Punctuation. I have corrected all I could find. 

Comment 5: Tanle. I have added a footnote.

Comment 6:Person first language. I have correct this.: 

Reviewer 2 Report

Manuscript #brainsci-280616 ‘Down Syndrome, Obesity, Alzheimer's Disease and

Cancer: A Brief Review and Hypothesis’ by D.W. Nixon is a serviceable review of the current understanding of the relationship(s) between DS, AD, and cancer, especially as it pertains to obesity and the hormones leptin and adiponectin. Although each of these topics is under research scrutiny, a combinatorial evaluation, both in clinical studies and basic modeling research, is warranted. The review is timely, although there are a few issues that require clarification.

1. BACE triplication. The author indicates BACE is triplicated on HSA21, which is only partially correct. Bace2 is triplicated in DS. However, Bace1, the principal beta-site APP cleaving enzyme that proteolyzes APP into Abeta and related APP metabolites, is not (it is on HSA11). Bace2 has not been shown to substantially cleave APP (it may do so under non-physiological conditions). The Reviewer suggests that ‘BACE’ description throughout the review be removed or revised accordingly.

2. Dementia in DS subjects. The percentage of aged DS subjects with (or without) dementia is currently being hotly debated. The author is urged to describe this issue in greater detail and indicate that diagnosing dementia in subjects with intellectual disability (such as DS) is challenging, especially as they age. Other methods and biomarkers, such as PiB PET scans, CSF measures of Abeta/tau, and emerging plasma screens that are currently being applied to DS subjects should be included in this section as well.

3. Associations (not causality) between leptin and adiponectin and AD. The author is urged to add a ‘limitations’ section on the findings of the review, especially as it pertains to the likely complex relationship(s) of both leptin and adiponectin to AD and cancer pathophysiology. There are literally dozens of bioactive compounds that change during the progression of AD but are not causally linked to the mechanisms underlying neurodegeneration or Abeta/tau pathology. An appropriate context of the current findings of these hormones in relation to AD is warranted. 

Author Response

Many thanks for the helpful critique. My responses are:

Comment 1: Bace. All BACE information is removed. 

Comment 2: Dementia in DS. My response is included in the new Limitations section.

 Comment 3: Issues with leptin and adiponectin are also in the new Limitations section (Section 5}.

Round 2

Reviewer 2 Report

The author addressed the critiques adequately.